# A New Angle Measurement in Translabial Ultrasound as an Adjunct for the Diagnosis of Pelvic Organ Prolapse

**DOI:** 10.3390/diagnostics12010098

**Published:** 2022-01-03

**Authors:** Gina Nam, Jae-Yen Song, Sa-Ra Lee

**Affiliations:** 1Department of Obstetrics and Gynecology, Chung-Ang University Hospital, Chung-Ang University College of Medicine, Seoul 06973, Korea; ginanam@caumc.or.kr; 2Department of Obstetrics and Gynecology, Seoul St. Mary’s Hospital, College of Medicine, The Catholic University of Korea, Seoul 06591, Korea; 3Seoul Asan Medical Center, Department of Obstetrics and Gynecology, University of Ulsan College of Medicine, 88, Olympic-ro 43-gil, Songpa-gu, Seoul 05505, Korea

**Keywords:** pelvic organ prolapse, translabial ultrasound, pelvic organ prolapse quantification (POP-Q)

## Abstract

The aim of this study was to compare the data obtained by a pelvic organ prolapse quantification (POP-Q) examination with the translabial ultrasound (TLUS) quantification of prolapse, using a new method of angle measurement. We analyzed the TLUS and POP-Q exam findings of 452 patients with symptoms of POP. The POP-Q system was used for clinical staging. TLUS was performed both at rest, and during the Valsalva maneuver after proper preparation. A horizontal reference line was drawn through the inferior margin of the symphysis pubis and the levator plate connected to the rectal ampulla, and the difference was calculated between the rest and the Valsalva maneuver. The Spearman’s correlation coefficient of agreement between the TLUS and the clinical POP-Q staging was used for statistical analysis. There was a weak degree of correlation between the POP-Q findings for the Ap parameter and our new angle measurement (rho = 0.17, *p* < 0.001). Thus, POP staging in conjunction with TLUS with this new angle measurement shows better agreement for the diagnosis of POP than POP-Q staging alone.

## 1. Introduction

Pelvic organ prolapse (POP) is a common disease among older women, wherein the organs in the female pelvis, including the uterus, bladder, and rectum, descend through the vagina. The prevalence of POP is increasing because of the aging of the global population. The lifetime risk of POP surgery is 12–19%, and more than 300,000 surgeries are performed annually in the United States [1].

To aid in the diagnosis of POP, the International Continence Society (ICS) introduced the POP-Quantification (POP-Q) system in 1996 to standardize the terminology of lower urinary tract function [2]. Gynecologists examine the patient’s pelvis and measure the degree of prolapse of the anterior wall, posterior wall, and cervix of the vagina or vaginal vault. However, this method can be subjective, depending on the examiner and the patient’s ability to perform the maximal Valsalva maneuver, and it can be inconvenient and unpleasant for the patient during the pelvic examination. In addition, the presence or absence of a rectocele and an enterocele is not clearly determined by the POP-Q stage [3]. Rarely, a vaginal mass mimicking prolapse can also be misdiagnosed as POP [4,5,6].

Although posterior vaginal prolapse has been considered a synonym of a rectocele, recent studies have shown that true rectoceles are only observed in 2.38–39% of women [3,4,5,6,7]. However, we cannot distinguish between simple posterior vaginal wall relaxation, a true rectocele, and an enterocele using only the POP-Q examination (Figure 1). Thus, the prevalence of true rectoceles and enteroceles remains largely unexplored.

Therefore, imaging techniques have been introduced to diagnose POP objectively and accurately, and to understand the pathophysiology of the underlying POP. Historically, evacuation proctography (EP), also known as a video defecography, is considered the “gold standard” for the diagnosis of rectoceles [8,9,10,11,12,13,14,15]. However, as EP is known to lead to the overdiagnosis of posterior compartment prolapse by identifying them in asymptomatic volunteers, and because it has significant interobserver variability, its results should be interpreted with caution [16,17,18]. Furthermore, EP requires ionizing radiation, bowel preparation, and it is embarrassing because the patient has to evacuate contrast in a nonprivate setting.

Translabial ultrasound (TLUS) has been used to assess the genitourinary tract and the degree of POP since the early 2000s [19,20,21,22]. Several studies have confirmed that TLUS is consistent, and its findings are in good agreement with those of video defecography [23,24,25,26]. A recent Cochrane review of imaging modalities for the detection of posterior pelvic floor disorders in women with obstructed defecation syndrome concluded that none of the imaging techniques met the criteria required to replace EP. Magnetic resonance imaging (MRI) and transperineal ultrasonography met the criteria of a triage test, with a positive test confirming the diagnosis of rectocele, enterocele, and intussusception, and a negative test ruling out a diagnosis of anismus [23].

However, there are only a few reports on the relationship between the POP-Q stage and TLUS findings [19,20,21,22].

Some authors emphasize the importance of the distance of the vertical lines on TLUS from a horizontal reference line through the inferoposterior margin of the symphysis pubis as an indicator of organ descent. However, the sonographic probe is slightly pressed against the labium, with an ample amount of jelly to obtain a good image, without the interfering effects of air. Hence, the distance of the vertical lines on TLUS from the horizontal reference line has limited accuracy in estimating the degree of POP because the organ protruding above the hymen level would be obstructed by the sonographic probe—a paradox of TLUS.

In the present study, we aimed to compare the data obtained by POP-Q examination and the TLUS quantification of prolapse. Our objective was to determine whether TLUS is clinically useful when compared to POP-Q for the diagnosis of POP. Thus, we expect that our method of angle measurement using TLUS will enable a more accurate diagnosis of POP.

## 2. Materials and Methods

### 2.1. Study Design and Patients

This retrospective study included 452 female patients who visited our outpatient clinic from April 2019 to September 2021 for symptoms of POP. We excluded patients with conditions mimicking POP. All patients were diagnosed with stage II or a higher degree of prolapse in at least one compartment, according to the POP-Q classification method. An expert urogynecologist (Dr. L.S.-R., with over 15 years of experience) examined the patients using the International Continence Society (ICS) POP-Q exam. The measured parameters were Aa, Ba, C, Ap, Bp, Gh, and Pb, as defined by the ICS POP-Q [2]. All the TLUS images were also routinely examined by the urogynecologist (Dr. L.S.-R.) and were interpreted using the angle measurement by the gynecologist (Dr. G.N.) to avoid interobserver variations. The TLUS was performed at rest and during the maximal Valsalva maneuver in the semi-Fowler position, with a transabdominal probe and a 3.5–5 MHz convex array transducer.

A rectocele was diagnosed as a discontinuity in the ventral contour of the anorectal muscularis. The depth of the rectocele was measured by the distance from the farthest point of the ampulla to the extended ventral line of the internal sphincter. An enterocele was diagnosed when the lower margin of the small bowel or omentum reached, or was below, the pubic bone. A true rectocele was defined as the presence of a discontinuity in the anterior contour of the internal anal sphincter and anterior anorectal muscularis, resulting in a diverticulum of the ampulla, indicative of a defect of the rectovaginal septum (RVS) [7,24,27,28].

We measured the angle between a reference line through the inferior margin of the symphysis pubis and the levator plate connected to the rectal ampulla, at rest and during the maximal Valsalva maneuver. A quantitative value was expressed as a difference in the angle—between the horizontal reference line through the inferior margin of the symphysis pubis and the levator plate connected to the rectal ampulla—between rest and the Valsalva maneuver (Figure 2). The difference in the angle estimated by the TLUS was compared with each measurement made using the POP-Q system.

### 2.2. Statistical Analysis

Continuous variables were expressed as mean ± standard deviation. We used Pearson’s correlation to assess the association between normally distributed continuous variables. A rho of 0 to 0.1 was classified as “very weak”, 0.1 to 0.3 as “weak”, 0.3 to 0.7 as “moderate”, and 0.7 to 1.0 as “strong”. *p* < 0.05 was considered statistically significant. Statistical analysis was performed using SPSS for Windows version 20 (SPSS Inc, Chicago, IL, USA).

## 3. Results

A total of 452 patients who visited our department for symptoms of POP were included in the analysis. The mean measurements of the ICS POP-Q coordinates are shown in Table 1. The data for the TLUS findings related to the angle difference are provided in Table 2. The mean angle difference was 17.56 ± 10.70 degrees. All parameters were normally distributed on Kolmogorov–Smirnov testing.

Table 3 displays correlations between the clinical and TLUS measures in all compartments. The results of the comparison of the quantitative angle measurements and the POP-Q parameters are provided. Aa, Ba, C, and Ap were weakly associated with angle differences (r = −0.11 for Aa vs. angle, −0.18 for Ba vs. angle, −0.14 for C vs. angle, and 0.17 for Ap vs. angle; *p* < 0.05 for all).

## 4. Discussion

In this study, we observed that the clinical staging and angle measurement were correlated with the POP-Q and TLUS findings. Weak but significant correlations were observed between the POP-Q stage and the angle measurements for the Ap parameter. Our results confirm that the angle between the rest and the Valsalva maneuver was significantly associated with the POP-Q stage, especially for the Ap parameter.

According to the POP-Q system, there are two anterior points (Aa and Ba), two posterior points (Ap and Bp), two apical points (C and D), a total vaginal length (tvl), a genital hiatus (gh), and a perineal body (pb) [2]. All these parameters are measured in centimeters, with the hymen as the reference plane. A POP-Q examination provides the quantitative criteria to obtain information on all the above parameters. The stages of POP have been described as stages 0 to IV, with 0 representing normal conditions, and III or IV describing a lack of support [2]. This system has demonstrated good interobserver and intraobserver reliability compared to other systems [29].

However, the five-level staging system of the current POP-Q system may be insufficient to describe POP accurately. In our cases, the POP-Q system revealed a posterior compartment prolapse, although it was a relaxation of the vaginal wall, not a rectocele or an enterocele by TLUS [3]. We also demonstrated a weak correlation between the POP-Q exam findings with the TLUS images for the presence of a rectocele and an enterocele [3]. The diagnosis of an anterior compartment prolapse can be aided with the addition of TLUS to a POP-Q exam. For example, a vaginal mass, such as a urethral diverticulum or a vaginal leiomyoma, can be a confusing factor that mimics POP [6]. Braga et al. [30] reported a patient with stage-II anterior vaginal prolapse using the POP-Q staging system, although the final diagnosis revealed a vaginal leiomyoma.

To date, several methods have been proposed to estimate POP using TLUS. The main purpose of TLUS imaging is the investigation of cystoceles, ureteroceles, enteroceles, and rectoceles, which can be misdiagnosed by using only the POP-Q system. Similar to the techniques used in defecography or dynamic MRI, the quantification of cystoceles, ureteroceles, enteroceles, and rectoceles has also been attempted with TLUS. Stress urinary incontinence was also assessed by the bladder neck position and mobility using TLUS [21]. Of all these parameters of hypermobility, bladder neck descent on TLUS was the only parameter associated with urodynamic stress incontinence.

Dietz et al. [19] were the first to report that TLUS can be used to quantify female POP. Measurements of TLUS were performed at rest and during the Valsalva maneuver. The inferior margin of the symphysis pubis was used as a reference line. To assess the maximum descent, the positions of the bladder neck or the leading edge of a cystocele for the anterior vaginal wall, the cervix (or the pouch of Douglas) for the central compartment, and the rectal ampulla for the posterior compartment were determined relative to the inferoposterior margin of the symphysis pubis. There was a good correlation for the anterior and central compartment prolapse between the ultrasound findings and the clinical staging. However, the poorest correlation was seen for the posterior vaginal wall prolapse between the ultrasound findings and the clinical staging (r = 0.53) [19].

Some authors found that the measurement of the anorectal junction position at rest, or the anorectal junction movement during straining, was simple and accurate with TLUS [22]. A true rectocele, due to the defect of the rectovaginal septum, can also be identified in the mid-sagittal plane [20]. TLUS reveals a true rectocele as a herniation of the rectal wall with a discontinuity in the anterior wall of the anorectum. The descent of the rectal wall through a defect on the anorectal wall on performing the Valsalva maneuver was quantified for the posterior compartment. Without any evidence of an actual fascial defect, a descent of the rectal ampulla below a reference line through the inferior symphyseal margin was calculated and defined as perineal hypermobility [20].

In many reports, the indicators of cystoceles, ureteroceles, and enteroceles on TLUS have been measured as the vertical distance to the most protruding part on the basis of the reference line of the lower margin of the symphysis pubis. The images of the TLUS at our hospital are comparable to those reported by previous studies. However, it is impossible to replace the POP-Q staging system with TLUS because the measurements of POP-Q are obtained with the effects of the Valsalva maneuver against the labium or perineum. Even if the examiner avoids exerting pressure on the perineum to allow the maximal Valsalva maneuver, the direction of the TLUS transducer is opposed to the pelvic organ descent. It is therefore not surprising that vertical distance measurements on the Valsalva maneuver, as observed in the previous studies, could be underestimated with TLUS rather than with POP-Q staging. Therefore, we defined an angle between a horizontal reference line through the inferior margin of the symphysis pubis and the levator plate connected to the rectal ampulla. Measurements were taken at rest and during the maximal Valsalva maneuver, and the difference was a numerical value used for comparing with the POP-Q system.

The decision on the type of surgery for each patient should be based not only on the POP staging, but also on the compartment of POP, age, sexual and social activity, occupation, and medical comorbidity [20,31]. The anesthesia and operation times for the specific type of surgery are also important for aged patients with comorbidities. Furthermore, we must consider the rates of recurrence and complications. Anterior compartment POP is the most troublesome type of POP, as its recurrence rate is the highest among anterior, apical, and posterior compartment POPs. In terms of the surgical treatment of an anterior compartment POP, a literature review concluded that a prosthetic treatment using mesh results in a higher anatomical success rate and less recurrence, while traditional anterior repair has fewer complications [31]. A recent randomized controlled trial comparing anterior colporrhaphy with a transvaginal trocar-guided mesh kit for cystocele repair also reported higher short-term success rates, but also higher rates of surgical complications and postoperative adverse events for the transvaginal trocar-guided mesh kit [32]. Therefore, the decision on the type of surgical treatment of POP should be individualized and based on the accurate diagnosis of POP, and with consideration to the advantages and disadvantages of the specific type of surgery.

This study has several strengths. First, to the best of our knowledge, this is the first study to suggest a new angle measurement method using TLUS to correlate with clinical POP-Q staging. Second, this study is retrospective, albeit with a relatively large number of patients. Third, the observations were based on the expertise of a single experienced urogynecologist, which minimized the interobserver reliability without any learning curve. Fourth, the anatomic abnormalities were assessed using TLUS, and the measurements and analyses were performed in a blinded manner by a pelvic ultrasound specialist (Dr. G.N), enhancing the reliability of the results. Finally, as the radiological imaging was performed in an outpatient clinic during a pelvic examination, it can be inferred that the patient performed the Valsalva maneuver correctly.

However, this study also has some limitations. First, this was a single-center retrospective study, and not a randomized controlled trial to test the effect of preoperative ultrasound for surgical planning and outcomes. Second, we did not correlate the clinical symptoms or surgical outcomes with TLUS findings. Third, we included all the POP patients, regardless of previous vaginal surgery, which cannot rule out the effect of scarring. However, in the cases of vaginal scar tissue, all the patients experienced symptoms of POP. Fourth, because a single gynecologist performed and measured the TLUS parameters in all cases, we cannot exclude the possibility of selection bias.

## 5. Conclusions

In conclusion, the staging of POP with the use of an adjunctive diagnostic TLUS using the levator plate as a reference line correlates more precisely with the results of a physical examination using the POP-Q system. Therefore, this angle measurement method can be used as an adjunctive parameter in TLUS imaging to correlate with the findings of the clinical POP-Q system, instead of the previous vertical line measurements.

## Figures and Tables

**Figure 1 diagnostics-12-00098-f001:**
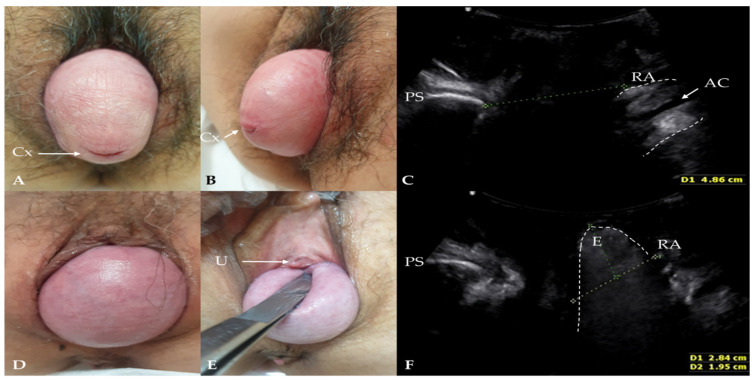
Clinical findings and translabial ultrasonography in patients with pelvic organ prolapse (POP) in the maximal Valsalva phase. (**A**–**C**) A patient with POP-Q stage IV without rectocele or enterocele. Cx: cervical os; AC: anal canal; PS: pubis symphysis; RA: rectal ampulla. (**D**–**F**) A patient with POP-Q stage III who underwent a hysterectomy. Enterocele was revealed in translabial ultrasonography during the maximal Valsalva maneuver. The contents of an enterocele appear generally iso- to hyperechogenic, and bowel peristalsis is usually observed in the enterocele sac. U: urethral meatus; E: enterocele.

**Figure 2 diagnostics-12-00098-f002:**
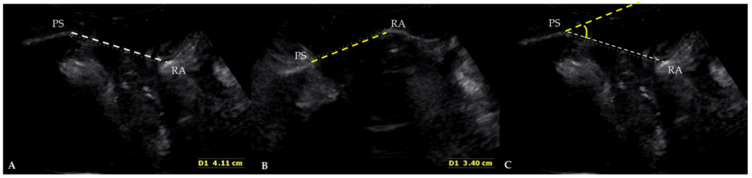
Translabial ultrasonography in patients with pelvic organ prolapse (POP) at rest (**A**), and in the maximal Valsalva phase (**B**). PS: pubis symphysis; RA: rectal ampulla. We measured the angle between a reference line through the inferior margin of the symphysis pubis and the levator plate connected to the rectal ampulla, at rest (white dotted line) and during the Valsalva maneuver (yellow dotted line) (**C**). A quantitative value was used to express the difference in the angle between a horizontal reference line through the inferior margin of the symphysis pubis and the levator plate connected to the rectal ampulla, between rest and during the Valsalva maneuver.

**Table 1 diagnostics-12-00098-t001:** Clinical findings of pelvic organ prolapse quantification (POP-Q) of 452 patients with symptoms of pelvic organ prolapse.

POP-Q Points (*n* = 452)	Mean ± SD	Range
Aa	0.94 ± 2.04	−3 to 3
Ba	2.86 ± 2.52	−3 to 10
C	0.41 ± 4.07	−9 to 10
Ap	−1.51 ± 2.65	−3 to 3
Bp	0.15 ± 3.02	−3 to 10
Gh	5.15 ± 1.38	3.5 to 8
Pb	3.46 ± 1.20	3 to 9

Aa: a point on the midline anterior vaginal wall 3 cm proximal to the hymen; Ba: maximum downward displacement of the anterior vaginal wall; C: maximum downward displacement of the cervix or vaginal vault; Ap: a point on the midline posterior vaginal wall 3 cm proximal to the hymen; Bp: maximum downward displacement of the posterior vaginal wall; Gh: length of the genital hiatus; Pb: length of the perineal body.

**Table 2 diagnostics-12-00098-t002:** Measurement of an angle on translabial ultrasound (TLUS) in 452 patients with symptoms of pelvic organ prolapse.

Parameter (*n* = 452)	Mean ± SD	Range
Angle difference (°) ^†^	17.56 ± 10.70	−18.14 to 74.79

^†^ The angle was measured between a horizontal reference line through the inferior margin of the symphysis pubis and the levator plate connected to the rectal ampulla, at rest and during the Valsalva maneuver.

**Table 3 diagnostics-12-00098-t003:** Correlation between translabial ultrasound (TLUS) and pelvic organ prolapse quantification (POP-Q) system (*n* = 452).

POP-Q Coordinate	Correlation Coefficient for Angle	*p*-Value
Aa	−0.11 ^†^	0.02
Ba	−0.18 ^†^	0.00
C	−0.14 ^†^	0.00
Ap	0.17 ^†^	0.00
Bp	−0.001 ^†^	0.98
Gh	−0.02 ^†^	0.68
Pb	−0.07 ^†^	0.12

Aa: a point on the midline anterior vaginal wall 3 cm proximal to the hymen; Ba: maximum downward displacement of the anterior vaginal wall; C: maximum downward displacement of the cervix or vaginal vault; Ap: a point on the midline posterior vaginal wall 3 cm proximal to the hymen; Bp: maximum downward displacement of the posterior vaginal wall; Gh: length of the genital hiatus; Pb: length of the perineal body. ^†^ Pearson’s correlation coefficient (rho).

## Data Availability

The Excel data used to support the findings of this study were supplied by Sa-Ra Lee under license, and requests for access to these data should be made to Sa-Ra Lee, leesr@amc.seoul.kr.

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
