# Peer review of "A New Angle Measurement in Translabial Ultrasound as an Adjunct for the Diagnosis of Pelvic Organ Prolapse"

_diagnostics, 2022, doi:10.3390/diagnostics12010098_

Round 1

Reviewer 1 Report

This is an ideal description of the rationale for having a sonographic technique to quantitatively measure the extent of clinical POP, and an ideal sufficiently sized observational case series to demonstrate its clinical benefit. I found no corrections that are needed to be made prior to its well-deserved publication.

Author Response

Letter to Reviewer from Author

Manuscript ID: diagnostics-1511142  

Title: A new angle measurement in translabial ultrasound as an ad-junct for the diagnosis of pelvic organ prolapse

We wish to thank the academic editor and reviewers for their thoughtful suggestions, which have improved our manuscript. I look forward to working with you and the reviewers to move this manuscript (diagnostics-1511142) closer to publication in the Journal of Diagnostics. The manuscript has been rechecked and the necessary changes made per the academic editor’s suggestions. Responses to all comments have been prepared and are shown in blue for your convenience. All the changes in the manuscript are highlighted in yellow.

(This manuscript got an English editing services)

 Thank you for your consideration. I look forward to hearing from you.

Reviewer 1:

Reviewer’s Comments and Suggestions for Authors:

This is an ideal description of the rationale for having a sonographic technique to quantitatively measure the extent of clinical POP, and an ideal sufficiently sized observational case series to demonstrate its clinical benefit. I found no corrections that are needed to be made prior to its well-deserved publication.

Response: Thank you for your motivating comments. We thank you for your kind, positive, and supportive comments.

Reviewer 2 Report

I read with great interest the manuscript, which falls within the aim of this Journal. In my honest opinion, the topic is interesting enough to attract the readers’ attention. Nevertheless, authors should clarify some points and improve the discussion, as suggested below.

Authors should consider the following recommendations:

  • Manuscript should be further revised in order to correct some typos and improve style.
  • Among POP, cystocele represents of the most challenging condition: the prolapse of anterior compartment, indeed, could be treated by both prosthetic surgery and native tissue repair. I suggest to discuss pro and cons of these two approaches, referring to: PMID: 26801794; PMID: 21561348.

Author Response

Letter to Reviewer from Author

Manuscript ID: diagnostics-1511142  

Title: A new angle measurement in translabial ultrasound as an ad-junct for the diagnosis of pelvic organ prolapse

We wish to thank the academic editor and reviewers for their thoughtful suggestions, which have improved our manuscript. I look forward to working with you and the reviewers to move this manuscript (diagnostics-1511142) closer to publication in the Journal of Diagnostics. The manuscript has been rechecked and the necessary changes made per the academic editor’s suggestions. Responses to all comments have been prepared and are shown in blue for your convenience. All the changes in the manuscript are highlighted in yellow.

(This manuscript got an English editing services)

 Thank you for your consideration. I look forward to hearing from you.

Reviewer 2:

I read with great interest the manuscript, which falls within the aim of this Journal. In my honest opinion, the topic is interesting enough to attract the readers’ attention. Nevertheless, authors should clarify some points and improve the discussion, as suggested below.

Authors should consider the following recommendations:

èResponse: Thank you for your motivating comments. We thank you for your kind, positive, and supportive comments.

Manuscript should be further revised in order to correct some typos and improve style.

  • Response: As per your comments, this revised manuscript got an English editing service once again and corrected some typos and inappropriate text style. We attached the proof of professional English editing.
  • Line 208-210.

However, the poorest correlation was seen for the posterior vaginal wall prolapse between ultrasound findings and clinical staging (r = 0.53) [19].

Among POP, cystocele represents of the most challenging condition: the prolapse of anterior compartment, indeed, could be treated by both prosthetic surgery and native tissue repair. I suggest to discuss pro and cons of these two approaches, referring to: PMID: 26801794; PMID: 21561348.

  • Response: We absolutely agree with your comment and most gynecologists are in agony about the high recurrence rate of anterior compartment POP. We added the following sentences in the discussion part with adding the relevant references (No. 31, 32).
  • Line 235-250.

The decision on the type of surgery for each patient should be based not only on the POP staging but also on the compartment of POP, age, sexual and social activity, occupation, and medical comorbidity [20,31]. The anesthesia and operation times for the specific type of surgery are also important for aged patients with comorbidities. Furthermore, we must consider the rates of recurrence and complications. Anterior compartment POP is the most troublesome type of POP as its recurrence rate is the highest among anterior, apical, and posterior compartment POP. In terms of surgical treatment of anterior compartment POP, a literature review concluded that a prosthetic treatment using mesh results in a higher anatomical success rate and less recurrence, while traditional anterior repair has fewer complications [31]. A recent, randomized controlled trial comparing anterior colporrhaphy with a transvaginal, trocar-guided mesh kit for cystocele repair also reported higher short-term success rates but also higher rates of surgical complications and postoperative adverse events for the transvaginal, trocar-guided mesh kit [32]. Therefore, the decision on the type of surgical treatment of POP should be individualized and based on the accurate diagnosis of POP, considering the advantages and disadvantages of the specific type of surgery.